# Multi-Objective Decision-Making Meets Dynamic Shortest Path: Challenges and Prospects †

**Juarez Machado da Silva** (ID)**, Gabriel de Oliveira Ramos** *(ID) **and Jorge Luis Victória Barbosa** (ID)

Applied Computing Graduate Program, Universidade do Vale do Rio dos Sinos, Av. Unisinos, 950, Cristo Rei, São Leopoldo 93022-000, Brazil; juarezmachado@edu.unisinos.br (J.M.d.S.); jbarbosa@unisinos.br (J.L.V.B.)
* Correspondence: gdoramos@unisinos.br
† This paper is an extended version of our paper published in the 2022 IEEE Congress on Evolutionary Computation (CEC), Padua, Italy, 18–23 July 2022.

**Abstract:** The Shortest Path (SP) problem resembles a variety of real-world situations where one needs to find paths between origins and destinations. A generalization of the SP is the Dynamic Shortest Path (DSP) problem, which also models changes in the graph at any time. When a graph changes, DSP algorithms partially recompute the paths while taking advantage of the previous computations. Although the DSP problem represents many real situations, it leaves out some fundamental aspects of decision-making. One of these aspects is the existence of multiple, potentially conflicting objectives that must be optimized simultaneously. Recently, we performed a first incursion on the so-called Multi-Objective Dynamic Shortest Path (MODSP), presenting the first algorithm able to take the MODM perspective into account when solving a DSP problem. In this paper, we go beyond and formally define the MODSP problem, thus establishing and clarifying it with respect to its simpler counterparts. In particular, we start with a brief overview of the related literature and then present a complete formalization of the MODSP problem class, highlighting its distinguishing features as compared to similar problems and representing their relationship through a novel taxonomy. This work also motivates the relevance of the MODSP problem by enumerating real-world scenarios that involve all its ingredients, such as multiple objectives and dynamically updated graph topologies. Finally, we discuss the challenges and open questions for this new class of shortest path problems, aiming at future work directions. We hope this work sheds light on the theme and contributes to leveraging relevant research on the topic.

**Keywords:** multi-objective; decision-making; shortest path; dynamic shortest path; multi-objective shortest path

## 1. Introduction

The Dynamic Shortest Path (DSP) problem plays an important role in many real-world situations in which the environment, typically represented as a graph, is subject to updates on its topology. DSP problems appear in a wide variety of application contexts and configurations, including logistics, telecommunications, and trip planning [1]. DSP algorithms become useful in several situations. For instance, in the case of trip planning, dynamic real-time traffic conditions (e.g., accidents) can be modeled as changes in the graph topology, which may be easier to handle than simply recomputing the desired path from scratch. However, the literature on DSP typically approaches the problem as having a single objective (e.g., travel time, in the case of trip planning), which does not always resemble how these problems are approached in the real world.

In scenarios where more than one objective needs to be considered when composing the solution, the relationship between these objectives needs to be carefully taken into account. In this scenario, the Multi-Objective Decision-Making (MODM) problem plays a fundamental role in searching for the best result, considering the trade-off between the

objectives and the preferences of the user or the system. MODM problems emerge in several real-life situations, such as the economic market, pathfinding, and project definitions [2]. However, applying MODM in the shortest path problem has been treated in the literature in the static form, also known as the Multi-Objective Shortest Path (MOSP) problem.

MOSP algorithms are unsuitable for solving the DSP problem, as any real-time update on the graph forwards the recalculation of the solution from scratch. Some works solve this problem by applying the MOSP with a time-dependent approach if the problem can be modeled with graph changes following a pattern. However, in a time-dependent MOSP problem, the variation in the values of the objectives is linked to time and conditions well-known a priori, such as rush hour on a highway. In this way, the time-dependent MOSP problem also does not allow real-time updates or unmapped conditions a priori without recomputing the entire solution from scratch. Therefore, the approaches to MOSP problems are not able to solve unpredictable changes in the graph.

Recently, we performed a first incursion on the so-called Multi-Objective Dynamic Shortest Path (MODSP) problem addressing the MODM bias to compose the cost of crossing edges in a DSP problem [3]. A MODSP problem is subject to unpredictable changes on the graph in real-time, and the edge costs comprise more than one objective. Since the MODSP handles DSP and MODM problems combined, an algorithm capable of tackling multiple objectives and avoiding recomputing the entire solution from scratch after an update is strategic.

In this article, we delved into the multi-objective dynamic shortest path problem by introducing its formal definition grounded on the DSP and MODM literature. In particular, we devised the MODSP as a unified view of these areas and elaborate on this new problem class in terms of its challenges, as well as its impact and prospects for the area. The main contributions of this work can be enumerated as: (i) a brief review of the DSP, MODM, and MOSP problems and the gaps in their methods to solve the MODSP problem; (ii) a complete formalization of the MODSP problem class and its relationship to the DSP, MOSP, SP, and MODM problem classes through a novel taxonomy; (iii) examples of different real-world applications as MODSP problem candidates. The MODSP problem's delimitation is formally presented for the first time with the intention of turning attention to this problem class.

To promote knowledge on the topic, we initially present an overview of the DSP in Section 2, MODM in Section 3, the MOSP in Section 4, preparing the reader for the formalization of the MODSP problem in Section 5. After formalizing the MODSP problem, Section 6 presents real-world application examples for the MODSP problem, followed by a discussion of the challenges and open questions in Section 7. Finally, we give the final considerations regarding this article in Section 8.

## 2. Dynamic Shortest Path

The DSP problem is a generalization of the Shortest Path (SP) problem, having applications in many areas like transportation networks, data flow analysis, database systems, and network routing [4–6]. Hence, to enhance comprehension, we begin with a formalization of the SP problem before introducing the DSP one. The SP problem can be represented by a graph $G = (V, E)$, where $V$ is a set of vertices and $E$ is a set of edges composed of the traversing cost. A path corresponds to a sequence of vertices connected by edges. In an SP problem, the goal is to find a path between a source vertex $s \in V$ and a destination vertex $d \in V$, whose cost (of the comprising edges) is minimum. Traditionally, finding the shortest path corresponds to finding the path with the shortest distance. However, to make this concept broader, here we refer to finding the route with the minimum cost (or optimal route), which better accommodates other cost definitions, such as distance, time, energy consumption, and comfort. Figure 1 shows an example of an SP graph where the shortest path regarding the edge costs is through the vertices $s \rightarrow c \rightarrow d$.

When an SP problem is subject to vertex insertion/deletion, edge insertion/deletion, or edge cost updates over time, the problem becomes a DSP problem [7]. This kind of

problem is present in real-life applications, such as changes in route planning, as well as changes in the routing of a computer network. A naive way to approach the DSP is to run an SP algorithm from scratch whenever the graph changes. However, this approach is inefficient since the knowledge obtained in previous executions is never reused [8]. The next subsection shows the existing classes of DSP problems in the literature [9].

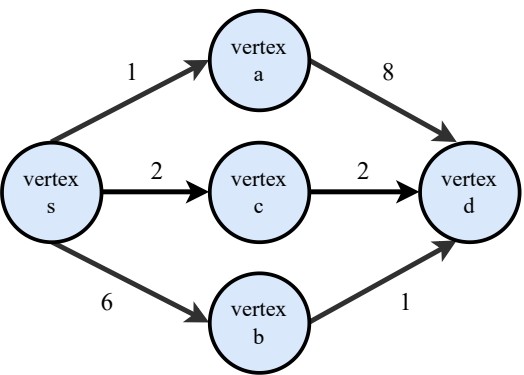

**Figure 1.** Shortest path graph example with vertices named by the letters $s, a, b, c$, and $d$ being $s$ the source vertex and d the destination vertex. The number over the edge represents the edge traverse cost between two vertices.

### 2.1. DSP Problem Classes

There are three classes of DSP problems that represent the dynamic characteristics of the problem [10]:

1. *Decremental dynamic shortest path:* a decremental DSP problem operates only with the deletion of vertices or edges.
2. *Incremental dynamic shortest path:* unlike the decremental DSP problem, the incremental DSP operates only with the insertion of edges or vertices.
3. *Fully dynamic shortest paths:* fully DSP problems allow edges and vertices' insertions and deletions.

Incremental and decremental dynamic shortest paths are sometimes referred to as being partially dynamic [11]. The three DSP problem classes are still subject to the basic SP problem structure that is applied to both dynamic and static problems. The following items organize the three basic structural points of an SP problem [12]:

1. *Directed or undirected graphs:* Directed graphs have specific directions in their edges. A directed edge from vertex $s$ to vertex $d$ can only be traversed from $s$ to $d$, but not from $d$ to $s$. In contrast, undirected graphs are those whose edges allow traversing from one vertex to another in any direction.
2. *Weighted or unweighted graphs:* Another factor that defines the structure of an SP problem is the cost to cross the edges of the graph. Weighted graphs are those with explicit costs $c(e)$ at every edge $e \in E$. Unweighted graphs have no costs at their edges, so a path's cost is defined by the number of hops (i.e., edges traversed) between the source vertex $s$ and the destination vertex $d$. Unweighted graphs can be seen as a specific case of weighted graphs where all the edge costs are equal, positive, and greater than zero.
3. *SP Problem Solution:* The desired solution is also linked to the problem structure and can be split into two main categories, Single-Source Shortest Path (SSSP) and All-Pair Shortest Path (APSP). The SSSP aims to find the optimal route from a single-source vertex to all other vertices on the graph. On the other hand, APSP aims to find the optimal route for every pair of vertices on the graph.

### 2.2. Representative Algorithms for DSP

This section presents some of the representative algorithms for DSP problems in the literature, pointing out at the end their limitation if applied to solve an MODSP problem.

The reader interested in the DSP state-of-the-art algorithms is referred to access the work of Henzinger [13] to be redirected to those works. Moreover, it is possible to find a literature review on traffic safety in Jiang et al. [14].

Algorithms addressed to solve the DSP problems in the literature can be exact or approximate. While the exact algorithms offer genuine responses regarding the optimal routes, the approximative algorithms propose solutions within a bound from the optimal [15]. The choice of which type of dynamic algorithm to use can be guided by the update and query time, which measure the efficiency of dynamic algorithms.

The exact algorithms present a polynomial amortized time cost for updates on the graph $\tilde{O}(n^k)$ with $k \approx 2$, where $n$ is the number of vertices [16]. The disruptive work for the fully dynamic APSP on directed graphs with non-negative weights was introduced by Demetrescu and Italiano [16]. Their work realized update operations in an amortized complexity of $O(n^2 \log^3 n)$ and each query in $O(1)$ worst-case time. Moreover, the local shortest path approach and the data structure adopted by Demetrescu and Italiano served as a base motivating more researchers to exploit the exact fully dynamic APSP algorithms. This was the case for Thorup [17], who made improvements on the Demetrescu and Italiano approach to the use of negative weights. Thorup's work reached a $O(n^2(\log n + \log^2(m/n)))$ amortized update time, where $m$ is the number of edges, maintaining the constant query time.

Approximate algorithms, despite not guaranteeing an optimal response, present responses within a bound of the optimal ($(1 + \epsilon)$-approximate [18]). Those algorithms have an amortized update time cost that is almost linear [19]. Roditty and Zwick [20] contributed to developing approximate algorithms for incremental and decremental SSSPs, which find the shortest path with the optimal solution proximity of $(1 + \epsilon)$ in an amortized complexity of $\tilde{O}(n)$. They also contributed with a full APSP that solves the updates with a time complexity of $\tilde{O}(mn/t)$, with $(1 \leq t \leq \sqrt{m})$, a query response time in $O(t)$. Brand and Nanongkai [18] also presented a fully APSP approximate algorithm using the fast matrix technique to improve the worst-case update time to $\tilde{O}(n^{2.045}/\epsilon^2)$; their work considered only directed graphs with positive weights at the edges. As a counterpart, the Brand and Nanongkai algorithm maintains only the distances and cannot report the corresponding paths.

In addition to those DSP algorithms, traffic networks are an important field in developing algorithms for the DSP problem, answering real-time traffic changes. Aiming to treat conditions such as accidents, traffic jams, and bad weather in real-time, Sever et al. [21] proposed a framework based on dynamic programming for hybrid routing policies in transit networks. Ardakani and Sun [22] developed an algorithm for the DSP problem considering the graph in continuous time to find the best route in real-time. Moreover, Fu and Rilett [23] proposed a way to find the shortest paths in traffic networks subject to dynamic and stochastic changes using a heuristic algorithm based on the K-shortest path.

The main drawback of DSP algorithms is the space complexity $O(mn)$ for storing the paths generated [16]. This disadvantage is necessary since the stored paths are the core that allows recalculating only the affected part by the update and avoids recomputing all paths from scratch. Now, if we look forward and try to solve an MODSP problem using an off-the-shelf DSP algorithm for each objective independently, the time and space complexity end up being elevated to a prohibitive level. Moreover, combining the optimal routes per objective into a single one is not trivial [24]. In this section, we covered the shortest path and dynamic shortest path scenarios as single-objective problems; the next section is dedicated to the MODM scenarios.

## 3. Multi-Objective Decision-Making

Multi-objective decision-making resembles many real-world problems. This is due to the fact that decisions are often weighted against more than one objective. When the objectives involved in decision-making are conflicting, one objective may be optimized at the expense of another one. In other words, there is a trade-off between objectives. In this

scenario, all trade-offs can be optimal solutions, and this condition turns the problem into a Multi-Objective Optimization Problem (MOOP). An MOOP is composed of $d$ objectives $\mathbf{c} = [c_1, c_2, \ldots, c_d]$ and $q$ decision variables $\mathbf{w} = [w_1, w_2, \ldots w_q]$, with $\chi = \{\mathbf{w} | \mathbf{w} \in \mathbb{R}^q\}$ denoting the decision space [25]. Equation (1) shows the vector of objective functions to be optimized for the MOOP resolution. The decision space $\chi$ regulates the objective vector $\mathbf{f}(\mathbf{w})$ with $\mathbf{f} : \chi \to \mathbb{R}^d$, where $\mathbb{R}^d$ represents the objective space [26].

$$\mathbf{c}(\mathbf{w}) = [c_1(\mathbf{w}), c_2(\mathbf{w}), \ldots, c_d(\mathbf{w})] \tag{1}$$

In multi-objective optimization, there are several possible solutions considering the trade-off between objectives. Comparing possible solutions is a way to indicate which ones stand out. Using the technique of comparing solutions, it is possible to look for a solution that is as satisfactory as any other considering most of the objectives, but strictly better in at least one of the objectives of the problem [27]. This solution is also called a non-dominated solution.

A set of non-dominated solutions in the objective space produces a boundary called the Pareto front. Figure 2 presents a Pareto front example with non-dominated solutions (also known as optimal solutions) and dominated solutions (also known as sub-optimal solutions). Analyzing Figure 2 from the objective $c_2$ perspective, solutions $c(\mathbf{w})$ and $c(\mathbf{w}')$ have the same value, i.e., $c_2(\mathbf{w}) = c_2(\mathbf{w}')$. Now, observing from the objective $c_1$ perspective, solution $c(\mathbf{w})$ is better than solution $c(\mathbf{w}')$, i.e., $c_1(\mathbf{w}) \prec c_1(\mathbf{w}')$. With both objectives $c_1$ and $c_2$ perspectives, the solution $c(\mathbf{w})$ weakly dominates the solution $c(\mathbf{w}')$. However, the $c(\mathbf{w})$ and $c(\mathbf{w}')$ solutions are not present in the Pareto front, making both sub-optimal solutions. Looking at the Pareto front, the $c(\mathbf{w}'')$ is an optimal solution since it dominates the solutions $c(\mathbf{w})$ and $c(\mathbf{w}')$.

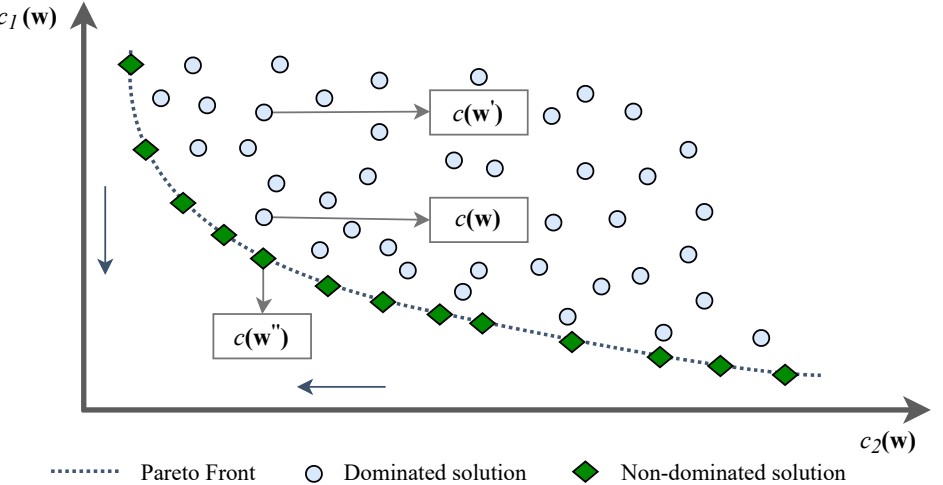

**Figure 2.** Bi-dimensional objective space with a Pareto front using standard nomenclature on the solutions.

An MOOP non-dominated solution enables an optimal choice for the decision-making process regarding the trade-offs between the objectives. Therefore, finding the Pareto front representing the best trade-off for the objectives solves an MOOP. However, computing an exact Pareto front can be unfeasible due to the huge number or the infinite number of non-dominated solutions [28]. Then, a Pareto front approximation is a good approach that requires less effort and can provide a set of non-dominated solutions accurately enough to represent the exact Pareto front [29]. With the Pareto front of the MOOP, it is possible to proceed with the decision-making process, selecting the optimal solution regarding the objectives' importance [24]. Then, an MODM problem is the composition of an MOOP and the decision-making process.

### 3.1. MODM Problem Classes

The purpose of MODM is to choose the optimal solution concerning the user's preferences. In this way, MODM problems can be organized into three distinct classes [27,30]:

1.  *Known weights:* In this scenario, the decision-making process already has its preferences in the form of weights that indicate the relative importance of each objective. This type of problem offers relatively small computational complexity, as it allows the direct search for the optimal Pareto solution. The scenario of known weights is usually handled with a priori methods, where a scalarization of the objectives can be determined before running the algorithm [30,31].
2.  *Unknown weights:* In this kind of problem, the objectives' weights are initially unknown due to the weights being subject to changes over time. In this case, a priori scalarization is unfeasible [27]. In this context, having an approximation of the Pareto front enables a quick answer when the weights' information is available. A solution to this problem is finding a set of solutions with at least one optimal solution for each set of weights. This set is known as the *coverage set*.
3.  *Support scenario:* The support scenario problem needs the user to interact with the system to indicate her/his preferences. As in the scenario with unknown weights, an approximation of the Pareto front must be found as a coverage set format. Afterwards, the user must indicate which of the optimal solutions available she/he prefers [27,31].

### 3.2. Representative Algorithms for MODM

Representative algorithms for MODM can be divided into three classes: classical mathematical-programming-based, evolutionary algorithms, and sequential decision-making algorithms with reinforcement learning. Mathematical-programming-based algorithms provide an optimal Pareto solution at each iteration. Two widely used classical methods are: the weighted sum approach [32,33] and the $\varepsilon$-constraint method [34,35]. A major drawback of these methods is that they need expert-level knowledge about the problem to define the objective weighting factor. Otherwise, there is no guarantee that the solution will be acceptable.

Evolutionary algorithms or Multi-Objective Evolutionary Algorithms (MOEAs) define a set of candidate solutions in each new iteration using a Pareto-based classification scheme. At the end of each iteration, those algorithms find a set of candidate solutions to compose the Pareto front. A strong point of using MOEAs to find the Pareto front is the capability of finding multiple Pareto-optimal solutions in just one simulation. However, MOEAs need attention on the algorithm choice and its parameterization since, quite often, the tuning of the parameters is necessary to reach the best results with reliability, robustness, and accuracy [36]. In addition to parameterization, there is also the issue of avoiding the local optimal solutions using fitness functions. Some of the most-important algorithms here include Non-dominated Sorting Genetic Algorithm II (NSGA-II) [37], Strength Pareto Evolutionary Algorithm 2 (SPEA-2) [38], the Multi-Objective Ant Colony System (MOACS) [39], and Multi-Objective Particle Swarm Optimization (MOPSO) [40].

Another form to address the multi-objective problems is using Sequential Decision-Making (SDM) algorithms, where the problem is modeled as a Markov Decision Process (MDP) [24]. An MDP can be defined as a tuple $\langle S, A, P, R, \gamma, \mu_o \rangle$, where:

*   $S$ is the state space, i.e., the set of all possible states in the environment;
*   $A$ represents the action space, i.e., the set of actions the agent can take;
*   $P$ specifies the transition probability function, which provides the probability of a state $s$ subjected to action $a$ at time step $t$ changing to the next state $s'$ at time $t + 1$;
*   $R$ is the reward function, specifying the reward signal obtained when performing action $a$ in state $s$ and transitioning to state $s'$;
*   $0 \leq \gamma < 1$ represents a discount factor, specifying the relative importance between future rewards and immediate rewards;
*   $\mu_o$ denotes the initial state distribution.

An SDM algorithm concerns the decision process in a given time step. This decision is dependenton the next states. Then, at each state, the SDM algorithm predicts the next best action in that state at that particular time step. SDM algorithms work with Reinforcement Learning (RL) and can produce globally optimal decision sequences, identifying the value of decisions with long-term perspectives [26].

When the model of the environment is not available a priori, the SDM with RL approach has the advantage of learning about the dynamics of the environment during the interactions. However, this approach also has drawbacks, since the mathematical and the evolutionary methods find complete solutions, while RL needs to build solutions interactively.

The disadvantage of MODM for dealing with an MODSP problem is its inefficiency in dealing with changing environments. In particular, the insertion/deletion of a graph's vertex or edge makes the problem non-stationary. If such a change occurs during a decision-making process, MODM will lose part of the knowledge acquired until that moment, and a systematic exploration may be necessary. With the SP, DSP, and MODM problems introduced until now, the following section presents the MOSP problem before the introduction to the MODSP for a better comprehension of the dynamic difference between both problems.

## 4. Multi-Objective Shortest Path

Some studies approach the MOSP as static graphs. Where most of these MOSP works are focused on bi-objective problems [41–45], a small part of the works address tri-objective problems [46–49], and few works approach many objectives [50–52]. A labeling technique extension, efficient for the single-objective purpose, is adopted by most part of these works to solve the MOSP problem [53]. Even using a generalization of the labeling technique to solve the MOSP problem, only one path indicating the best value for all criteria is not available. Therefore, solving an MOSP problem is fundamental to search for non-dominated paths in the graph (Pareto front of the MOSP).

The labeling technique applied to the MOSP is iterative and employs the method of computing the one-to-all shortest paths, a single-source solution for the MOSP problem. We can divide the labeling technique into two: label-setting and label-correcting. Each of those labeling techniques has its own update procedure to reach the final optimal shortest paths. The key difference between both is in the estimation of the distance to the shortest path under evaluation regarding each vertex in each iteration.

For the label-setting algorithms, the vertex under evaluation searches for the least-possible label value. Then, if the optimality principle holds, each vertex is scanned at most once to reach the optimal shortest path from the source vertex to the destination vertex [53]. This means that, in an MOSP problem, where the goal is a one-to-one optimal shortest path, the label-setting method finishes the search when it reaches the destination vertex, avoiding a more costly search in all graph vertices.

In contrast, a label-correcting algorithm may select a vertex more than once and ends after finding the shortest path from the source vertex $s$ to all other vertices $i \in V$. This effect occurs since the label-correcting algorithm follows a policy to scan the graph's vertices, for instance First In, First Out (FIFO). Moreover, the shortest path distance computed at each interaction is temporary and converges to the one-for-all optimal shortest path distances. Therefore, this scan method of the label-correcting algorithm makes the computation time to find the shortest one-to-all paths the same as that of the shortest one-to-one path.

An MOSP problem may also consider a dynamic change over time (time-dependency) regarding the objectives' cost over the edges [54]. This time-dependent approach to the MOSP problem considers the time window in a vertex under evaluation and the time spent to reach the neighbor vertices regarding the objectives' cost in the current window time. In this way, a time-dependent algorithm assesses the objectives' cost over time for each vertex under evaluation until it reaches the destination vertex [55]. This approach is appropriate

for predictable changes in graphs over time, which is applicable, for example, to determine the shortest path in a road map regarding the traffic prediction in the length of the day.

Another way to approach the MOSP problem is by using speedup techniques to search for the optimal shortest paths. This approach has a preprocessing phase that considers the objectives' costs and a query phase to search for the optimal shortest paths. The preprocessing phase sets the Pareto-optimal paths as the most costly phase. In contrast, the query phase is remarkably fast, receiving as inputs the source vertex, destiny vertex, and weights unknown a priori to search for the shortest path with the best trade-off between the objectives' cost and the user preferences [56]. Geisberger et al. [57] classified speedup methods into three classes:

1. *Hierarchical methods:* As the name implies, this method extracts the dependent hierarchy information from the graph. The hierarchical method most used is the Contraction Hierarchies (CHs) proposed by Geisberger et al. [58]. The CHs speedup technique performs the ordering of the vertices employing vertices' contraction in the preprocessing phase. Vertex contraction consists of adding shortcut edges in relation to the graph's contracted vertices, thus reducing the search space for the query phase. In the query phase, the search for shortest paths using modified bi-directional Dijkstra algorithm can respond to queries quickly due to the contraction of vertices performed in the preprocessing phase.
2. *Goal-directed methods:* This method avoids scanning the vertices not able to reach the destination vertex exploiting the graph structure. The most-used speedup technique is a goal-directed one, the A* with Landmark and Triangular inequality (ALT) proposed by Goldberg and Harrelson [59]. In this speedup technique, the preprocessing phase consists of identifying a small number of reference points for a given set of vertices concerning all other vertices of the graph. Then, in the query phase, the destination vertex is provided, and the triangular inequality produces two lower bounds that allow the search to be performed with a sense of direction.
3. *Method combinations:* A combination of speedup methods is used to improve the speedup taking advantage of the individual graph properties of each method involved in the combination.

Bast et al. [60] thoroughly surveyed the speedup methods and their applications. Speedup methods are widely used in route planning on road networks without considering unpredictable updates in the graph structure [57,61]. For example, supposing that a new vertex needs to be inserted in the road network, the preprocessing phase needs to be rerun from scratch, not taking advantage of the previous preprocessing solution.

*Representative Algorithms for MOSP*

In this section, we describe some representative algorithms for MOSP problems, which are the starting point of our research and guide us to the MODSP problem definition. Considering the reader interested in a deeper study of MOSP problems, a good starting point is the works from Bast et al. [60] for speedup techniques and Zajac and Huber [62] for applications using labeling techniques and time-dependent techniques.

Regarding the label-setting algorithms, Mandow and De La Cruz [63] proposed a New Approach to Multi-Objective A* (NAMOA*). The algorithm provides exact results with lower-bound estimates exploring an optimal number of labels. However, tests and results are not present in the paper. After that, a dimensionality reduction technique for the NAMOA* algorithm was proposed by Pulido and Mandow [50]. The work reached the reduction of labeling processing time using the optimized dominance checks approach. Their work was tested with the New York road network with three, four, and five objectives. Finally, they achieved the best results, considering the selection order for the open labels for three different configurations of the NAMOA* algorithm.

Regarding label correction algorithms, Skriver and Andersen [64] proposed an algorithm based on the work of Tung and Chew [65], considered efficient until the present for MOSP problems. Their algorithm works in a reverse way initializing the label informa-

tion from the destination vertex and finding in a fast way to the shortest paths from an intermediate vertex to the destination one.

The work by Bökler and Mutzel [66] pointed out that the label-setting approach does not perform well when problems are composed of more than two objectives. Due to this, their work focused on label-correcting, presenting good results for real-world instances.

Considering the time-dependent MOSP, Maristany de las Casas et al. [67] introduced an algorithm able to tackle predictable dynamic changes. Their algorithm uses a Fully Polynomial-Time Approximation Scheme (FPTAS) for finding MOSP approximate solutions in $O((n \log n + m)T)$ runtime, where $T$ is the tensor size that stores the number of paths in each vertex. However, their approach does not accept a posteriori updates on the graph structure. This means that the algorithm proposed by Maristany de las Casas et al. cannot solve MODSP problems without avoiding recomputation from scratch when unpredictable changes come.

Looking at the speedup approaches, Funke and Storandt [56] presented a CHs algorithm that supports weight changes to determine the trade-off between the multi-objective costs in the query phase. Geisberger et al. [57] presented an example of a combination of the CHs and ALT methods for route planning with flexible objective functions. However, the design of those speedup algorithms is for the MOSP problem. Then, if there is any change in the graph structure regarding objectives' costs or the insertion of a new unforeseen vertex, the pre-processing phase needs to be performed from scratch.

After presenting the perspectives of the DSP and MODM problems, as well as the difference between the MOSP and MODSP problems, the next section is entirely dedicated to the description of the MODSP problem.

## 5. Multi-Objective Dynamic Shortest Path

Given the current panorama of the DSP and MODM research, we now turn our attention to the intersection of these two problems. This intersection results in the proposed MODSP problem as a generalization of the DSP problem with multiple objectives. The MODSP problem has multiple objectives present in the composition of the costs to traverse the graph and allows the update of the objectives' costs at any time. Moreover, the MODSP problem involves finding the optimal path regarding the objectives between two vertices in the graph. Last but not least, the MODSP problem allows the insertion and deletion of graph edges and vertices at any time.

To better understand the MODSP problem, we start with the definition of the problem itself, followed by a taxonomy for a better view of the problem considering the MODSP and MODM problem classes and their subclasses. Finally, we present some application examples for the real world.

### 5.1. Problem Definition

The Multi-Objective Dynamic Shortest Path (MODSP) problem is characterized as a graph $G = (V, E)$, where $V$ is the set of vertices and $E$ is the set of edges of the graph. An edge $e \in E$ has its traversing cost $\mathbf{c}_e$ addressed by $d$ objectives $\mathbf{c}_e = [c_{e1}, c_{e2}, \ldots, c_{ed}]$. The trade-off between the objectives is regulated by the decision variable vector $\mathbf{w}$ composed of $q$ decision variables $\mathbf{w} = [w_1, w_2, \ldots, w_q]$, resulting in Equation (1). Then, with the cost vector $\mathbf{c}_e$ (from an edge or a path) and the decision variable vector $\mathbf{w}$, it is possible to designate a linear scalarization function and compute the inner product from both vectors $c_e(\mathbf{w}) = \mathbf{c}_e \cdot \mathbf{w}$.

We can express the vertex sequence of a path $\pi_{xy}$ from vertex $x$ to vertex $y$ by $\pi_{xy} = \langle x_0, x_1, \ldots, x_k \rangle$, $x_0 = x$, $x_k = y$, and $(x_i, x_{i+1}) \in E$ for each $i$, $0 \leq i < k$. Observing the edge $(u, v) \in E$ from Figure 3, its multi-objective cost to traverse the vertex $u$ to $v$ is expressed by $\mathbf{c}_{uv}$. Moreover, we can denote the multi-objective cost of a path $\pi_{xy} \in G$ by $\mathbf{c}_{\pi_{xy}} = \sum_{i=0}^{k-1} \mathbf{c}_{x_i x_{i+1}}$ and its linear scalarization regarding the decision variables by $c_{\pi_{xy}}(\mathbf{w}) = \mathbf{c}(\pi_{xy}) \cdot \mathbf{w}$.

A special case in modeling the MODSP problem is a path from a vertex $x$ to itself $\pi_{xx} = \langle x \rangle$, where all the objectives' costs are equal to zero. Observing Figure 3, a concatenation between the paths $\pi_{xv} = \langle x, \ldots, x', v \rangle$ and $\pi_{vy} = \langle v, y', \ldots, y \rangle$ can be written as $\pi_{xv} \cdot \pi_{vy} = \langle x, \ldots, x', v, y', \ldots, y \rangle$. Finally, we denote the path $\pi_{xb}$ as the left subpath $\ell(\pi_{xy})$ to compose the path $\pi_{xy} = \pi_{xb} \cdot \langle b, y \rangle$ in the same way and the path $\pi_{ay}$ as the right subpath $r(\pi_{xy})$ to compose the path $\pi_{xy} = \langle x, a \rangle \cdot \pi_{ay}$.

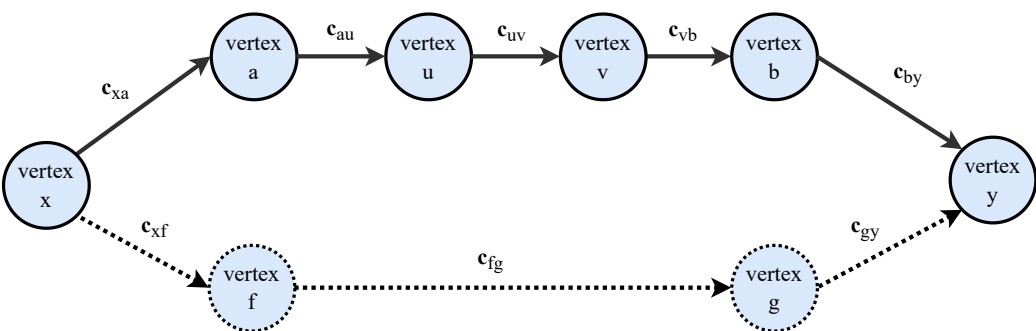

**Figure 3.** Multi-objective dynamic shortest path graph example where the dashed lines represent vertices and edges added at an unpredictable moment. The vertices are named by the letters $x, a, u, v, b, f, g$, and $y$, being $x$ the source vertex and y the destination vertex. Vector **c** over the edge represents the edge traverse multi-objective cost between two vertices.

We can observe two moments for choosing a path $\pi_{xy}$ in Figure 3. In the first moment, assume that vertices $f$ and $g$ do not exist in the graph. In this case, the possible path is through vertices $a, u, v, b$. Choosing this unique available path results in a cost denoted by $\mathbf{c}_{xa}(\mathbf{w}) + \mathbf{c}_{au}(\mathbf{w}) + \mathbf{c}_{uv}(\mathbf{w}) + \mathbf{c}_{vb}(\mathbf{w}) + \mathbf{c}_{by}(\mathbf{w})$, which can be expressed as $\pi_{xv} \cdot \pi_{vy}$. In the second moment, assuming new vertices $f$ and $g$ are inserted in the graph, it is necessary to consider the newly available path through the new vertices verifying if the solution is non-dominated. In this case, the new path has a cost denoted by $\mathbf{c}_{xf}(\mathbf{w}) + \mathbf{c}_{fg}(\mathbf{w}) + \mathbf{c}_{gy}(\mathbf{w})$ or $\pi_{xg} \cdot \pi_{gy}$.

As we mentioned before, an MODSP problem is subjected to an unforeseen sequence of vertex updates. To maintain these properties, the MODSP problem is subject to the following operations:

- update $(v, E_v, \mathbf{c}')$: This procedure allows dynamic changes in the graph structure, updating, inserting, or removing the vertex $v$ and the cost vector $\mathbf{c}'$ of all edges $E_v$ connected to the vertex $v$. Furthermore, this procedure updates only the paths that cross the vertex $v$, taking advantage of the previous solution not affected by the update;
- pf_sp $(x, y)$: a procedure that returns the approximate Pareto front of optimal shortest paths from vertex $x$ to vertex $y$, if any;
- cost_sp $(x, y, \mathbf{w}')$: a query procedure that receives as the input the source vertex $x$, the destination vertex $y$, and the decision variables 'vector $\mathbf{w}'$, returning optimal cost vector $\mathbf{c}_{xy}$ from the optimal shortest path, if any;
- optimal_sp $(x, y, \mathbf{w}')$: This query procedure uses the inputs' source vertex $x$, destination vertex $y$, and the decision variables' vector $\mathbf{w}'$ to return the optimal shortest path $\pi_{xy}$ considering the best trade-off between the decision variables' vector $\mathbf{w}'$ and the cost vector $\mathbf{c}_{xy}$, if any.

Bringing Figure 2 to the MODSP context, we can have several possible solutions considering the trade-off between the objectives for only one desired path $\pi_{xy} \in G$. Figure 4 shows the same example of the Pareto front of Figure 2, in which we changed the solutions' nomenclature to possible paths to facilitate the understanding of the proposed problem. Next, we rewrite the description given in Section 3 for the context of the MODSP problem. In Figure 4, when comparing the solutions $c_{\pi_{xy}}(\mathbf{w})$ and $c_{\pi_{xy}}(\mathbf{w}')$, considering only the objective $c_{\pi_{xy}1}$, we have the solution $c_{\pi_{xy}1}(\mathbf{w})$ as dominant, $c_{\pi_{xy}1}(\mathbf{w}) \prec c_{\pi_{xy}1}(\mathbf{w}')$. However, when considering the goal $c_{\pi_{xy}2}$, both solutions have the same value, $c_{\pi_{xy}2}(\mathbf{w}) = c_{\pi_{xy}2}(\mathbf{w}')$.

Thus, the solution $c_{\pi_{xy}}(\mathbf{w})$ weakly dominates the solution $c_{\pi_{xy}}(\mathbf{w}')$, and both solutions are sub-optimal due to being out of the Pareto front. Finally, the $c_{\pi_{xy}}(\mathbf{w}'')$ solution is an example of a non-dominated solution present on the Pareto front for a specific trade-off defined for $\mathbf{w}$.

The set of non-dominated solutions of a path $\pi_{xy} \in G$ contains the best trade-offs for the multiple objectives. Thus, the MODSP solution regards maintaining the sets of non-dominated shortest paths up to date after each vertex update operation, taking advantage of previously computed solutions. In this way, the decision-making process can proceed with the query procedures.

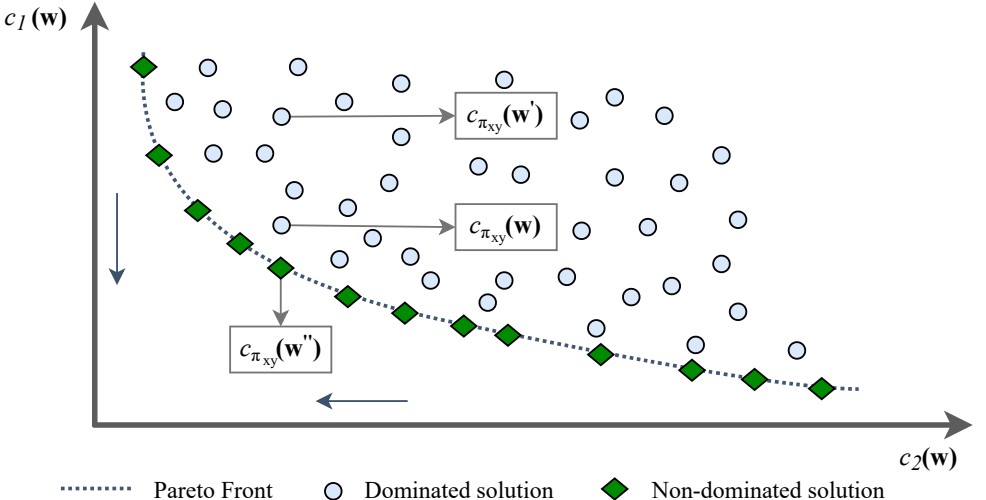

**Figure 4.** Bi-dimensional objective space with the Pareto front using the path nomenclature on the solutions.

*5.2. MODSP Taxonomy*

In order to make the MODSP problem more visible as a class of problems, Figure 5 shows a taxonomy, with our view of the problem classes that deal with the shortest path problems. Starting the taxonomy analysis by segregating the five classes involved in the Venn diagram, we can observe that the three problem classes, the MODSP, DSP, and MODM, are well delimited in the taxonomy. We represent the MOSP problem class in the taxonomy as the intersection between the MODM and MODSP problem classes. Our claim that the MOSP is the intersection between MODM and the MODSP implies that the MOSP definition can neither represent dynamic structural changes in the graph over time nor represent graph-less problems with multiple objectives. In the same way, the SP problem class is presented in the taxonomy as an intersection area between the DSP and MOSP class of problems. As the SP problem is a subclass of the DSP and MOSP class of problems, the SP problem class does not have the multiple objectives from the MOSP problem class and does not have the dynamic graph structural changes over time from the DSP problem class.

By analyzing the taxonomy, it is possible to observe that the SP and MOSP problems are subclasses of the MODM since they can be solved using multi-objective reinforcement learning methods [30]. Furthermore, the MODM problem class can solve other non-graph-related problems, as represented in Figure 5, by the area that does not intersect with the MODSP problem class. However, the MODM problem class cannot solve graph problems with dynamic changes like the MODSP and DSP problems due to the possibility of inserting or removing vertices and edges at any time [1,12]. Taking the analysis to the bias of the MODSP problem class, it is possible to observe that the MODSP methods can solve all problems related to graphs since the DSP, MOSP, and SP problem classes are subclasses of the MODSP problem class. Our taxonomy of the MODSP problem class sheds light on a challenging problem in emerging areas, such as the evolution of telecommunications networks and autonomous vehicles. These emerging areas make MODSP more evident

in real-world situations, requiring attention to be resolved appropriately and avoiding delayed and inaccurate responses.

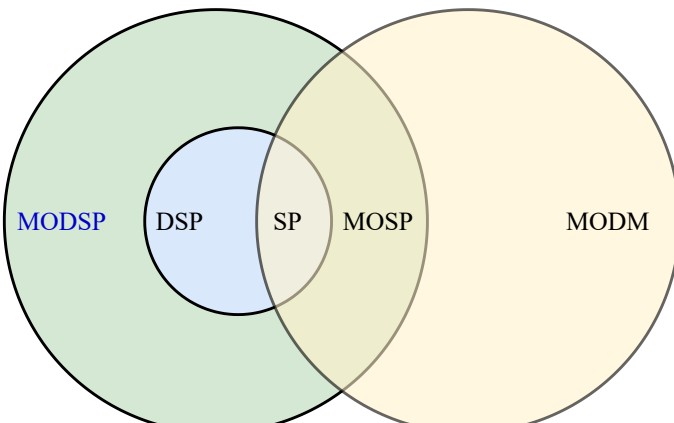

**Figure 5.** Venn diagram representing the taxonomy of the MODSP problem class and its relationship with the DSP, MOSP, SP, and MODM problem classes.

To the best of our knowledge, until our previous work [3], no others in the literature dealt completely with MODSP. The closest works considered only MOSP approaches [41–43,46–48,50,51]. The MODSP distinguishes itself from the MOSP since it requires a solution or an update after an online structural change on the graph. Due to the absence of the dynamic and unpredictable update component in the problem formulation, the MOSP needs to recalculate the shortest paths from scratch when the insertion or deletion of a route in the graph occurs. However, recalculating optimal routes from scratch is inefficient since changes in the graph may occur at exclusive points. Thus, in the MODSP algorithmic perspective, the graph changes are evaluated, and the update is executed only on the affected paths. Concerning the problem's complexity, the MOSP is NP-hard [68,69], which also makes the MODSP at least NP-hard.

## 6. Applications

The list of potential applications of the MODSP is increasing steadily, especially with the advent of smart cities, autonomous vehicles, smart devices, the Internet of Things, etc. Furthermore, access to mobile networks allows smart devices to benefit from the MODSP when choosing and updating a displacement. At the same time, the MODSP applications can still vary between leisure and strategic decisions. Therefore, this section presents a few selected strategic scenarios for the application of the MODSP.

### 6.1. Intelligent Transportation Systems

Traffic networks comprise a problem addressed to the transportation systems that involves numerous variables to be satisfied, e.g., time travel, distance, safety (drainage problems, weather, and lack of illumination), and travel costs [70–72]. All those variables can be updated dynamically with a real-time system, as indicated in the work of Sohrabi and Lord [73]. Therefore, we see the traffic networks as an MODSP problem since the problem can receive real-time updates regarding the objectives cost and changes in the graph topology due to incidents that can close or open roads at any time. Moreover, the optimal shortest path result will be subjected to user preferences where a user selects a fast travel time rather than driving a short distance.

Works addressed to transportation systems consider the traffic networks dynamic when the weight changes over time or the objectives' cost follows a pattern. However, as Sohrabi and Lord [73] listed in their work, the desired requirements for a safe-route-finding system regard the capability to update the graph information in real-time with the support of sensors and dealing with trade-offs over the travel time. Then, following this future of

the traffic network problem, the MODSP problem fits as a better model for developing new solutions.

Moreover, another current traffic network problem that calls our attention is the navigation system used by self-driving cars. In such a problem, vehicles are also subject to dynamic environmental changes. Therefore, they need to accommodate multiple objectives regarding the optimal route choice, such as travel time, comfort, energy consumption, obstacles, distance to be covered, and threats to passengers and pedestrians [60,73–75].

### 6.2. Military Path-Finding

Military path-finding is a problem for the chain of command and control in carrying out military missions. Military missions are considered highly dynamic and uncertain when involving human lives. Military path-finding is *dynamic* due to the nature of the battlefield, where the terrain and enemies can change at any time. Moreover, the missions are *uncertain* factors due to the multiple objectives present in the mission, such as the visibility of the terrain, distance, difficulty in accessing the route, execution time, cost operations, and exposure to threats.

Some works address the problem with a focus only on the multi-objective issue, leaving the displacement route in enemy territory undefined [76]. More comprehensive works deal with displacement and the multiple objectives involved in the mission [77,78]. However, these works do not consider route insertion or deletion when troops traverse the path. Considering existing results for the military path-finding problem, a complete MODSP approach remains an open problem.

### 6.3. Routing Protocols

Internet routing has a dynamic nature because a link may change its status to up or down during the transmission of the packets. Besides that, routing protocols are MODM problems as well [79–82], where the objectives may be listed as:

- *Maximizing bandwidth*, thus allowing for rapid transfer of data between nodes;
- *Minimizing latency*, to keep buffering low during distributed real-time applications;
- *Maximizing redundancy*, then providing multiple data paths so that connectivity is preserved if some nodes go offline.

Routing protocol challenges are present in streaming platforms with a high bandwidth, as well as real-time applications such as video calls requesting low latency. Given these requirements, routing can be modeled as an MODSP problem, where the system can dynamically redirect traffic flow to achieve better results.

### 6.4. Unmanned Aerial Vehicles

Unmanned Aerial Vehicles (UAVs)/drones are increasingly used in many activities, reducing life-threatening conditions or in environments with restrictive access. For example, UAVs can deliver packages, film live sports, and search for people for diverse purposes [83–85]. For these applications, UAVs should consider other obstacles along the path, such as other drones or even trees and traffic signs; moreover, they should also reduce energy consumption to cover larger areas [86]. To this end, the UAVs fit the definition of the MODSP to accomplish the dynamic changes in the environment and multiple objectives' cost.

## 7. Challenges and Open Questions

Recent years have seen a relevant breakthrough regarding multi-objective and shortest-path algorithms due to complex real-world problems. However, as presented in Section 5, this work opens a new research niche for the shortest-path problems unifying the dynamic and multi-objective approaches resulting in the MODSP problem. Since the MODSP is a new problem proposal, a number of significant and pressing challenges for the MODSP research need to be intensified. In this context, the remainder of this section will present an overview of these challenges.

### 7.1. Lack of Multi-Objective Dynamic Shortest Path Datasets

The datasets are essential to perform tests and develop algorithms capable of solving the MODSP problem. The proposal to adopt the MOSP dataset made by da Silva et al. [3], using the initial dataset to build the graph update datasets, is valid for testing. In the approach by da Silva et al., the creation of the graph update dataset starts with choosing a random vertex and all edges that connect with the selected vertex generating the new dataset for update purposes. However, this type of approach may not be sufficient a priori, as the datasets for updating may be presented in a more complex way and correlated with real-life problems. The datasets for updating should also cover the insertion of new vertices not present in the graph's initial form, and the removal of vertices from the graph enable this. This challenge of finding or building datasets can slow down and even hinder researchers' progress on the MODSP problem. Then, actions towards this challenge are related to the disposal of MODSP problems, data, benchmarks, and baseline implementations available on open platforms.

### 7.2. Lack of Algorithmic Benchmarks

Given the novelty of the MODSP problem, there is a lack of possible solutions, bringing to light the need to engage more researchers in the area. This challenge is directly related to helping other researchers interested in the MODSP. One possibility is the creation of a repository of algorithms following a unified standard in order to facilitate the benchmark of new methods that may be developed in the future. In this way, newly proposed algorithms to solve the MODSP problem will emerge and allow an adequate benchmark between them. Such engagement of more researchers could bring benefits to our society in solving the MODSP problem and taking it to applications in real-world problems.

### 7.3. Algorithmic Analysis for a Better Evaluation of the MODSP Solutions

Even with the lack of algorithms to solve the MODSP problem, the question of the true advantage of using such solutions remains open. The challenge for this type of questioning is finding an answer in the analysis from the theoretical point of view through asymptotic analysis. One piece of evidence of needing a theoretical analysis of existing algorithms was presented in the work by da Silva et al., where it was clear that the cost of generating paths at the first moment is very high, surpassing an MOSP algorithm. However, the cost of updating in a second moment has a significant advantage over other algorithms. In this way, it is evident that an asymptotic analysis of the MODSP algorithms is necessary to evaluate in what circumstances or situations there are advantages in using these algorithms about the others of the DSP or MOSP considering an MODSP problem for the resolution.

## 8. Concluding Remarks

This article generalized the dynamic shortest path problem, which determines routes in graphs subject to the insertion, deletion, and update of edges and vertices, where the cost of crossing the edges comprises multiple objectives subjected to decision variables unknown at first. We refer to this problem as the Multi-Objective Dynamic Shortest Path (MODSP) problem since it lies at the intersection of two widely explored problems, namely the Dynamic Shortest Path (DSP) and Multi-Objective Decision Making (MODM). To this end, we described the DSP and MODM problem classes focused on the works that inspired the proposal of the MODSP problem. Moreover, we also described the Multi-Objective Shortest Path (MOSP) problem and the time-dependent MOSP problem evidencing the differences with the MODSP problem.

Furthermore, this article described the theoretical formulation of the MODSP problem by presenting a taxonomy in Venn diagram format. The taxonomy introduced the MODSP problem class and its relationship with the MODM, DSP, MOSP, and SP problem classes. Considering that the MODSP problem covers many complex situations in the real world, some applications were suggested so that the problems bring motivation to be treated more

thoroughly in future works. In addition, we provided our current view of the challenges and open questions for the MODSP problem.

More than formalizing or delimiting the MODSP problem, this work aimed to draw researchers' attention to this relevant class of problems. Considering current technologies, such as sensors and devices connected in the cloud, we reach real-time conditions reflecting route systems much more dynamically than years ago. Therefore, we believe the MODSP problem cannot be neglected currently due to its great practical importance, as many applications depend on this approach to find satisfactory solutions.

**Author Contributions:** Conceptualization, J.M.d.S.; investigation, J.M.d.S.; methodology, J.M.d.S.; supervision, G.d.O.R. and J.L.V.B.; writing—original draft, J.M.d.S.; writing—review and editing, G.d.O.R. and J.L.V.B. All authors have read and agreed to the published version of the manuscript.

**Funding:** This research was partially supported by Coordenação de Aperfeiçoamento de Pessoal de Nível Superior, Brasil (CAPES), Finance Code 001, Conselho Nacional de Desenvolvimento Científico e Tecnológico (CNPq) (grant 306395/2017-7), Fundação de Amparo à Pesquisa do Estado do Rio Grande do Sul (FAPERGS) (Grants 17/2551-0001197-0 and 19/2551-0001277-2), and Fundação de Amparo à Pesquisa do Estado de São Paulo (FAPESP) (Grant 2020/05165-1).

**Institutional Review Board Statement:** Not applicable.

**Informed Consent Statement:** Not applicable.

**Data Availability Statement:** Not applicable.

**Acknowledgments:** This research was partially supported by Coordenação de Aperfeiçoamento de Pessoal de Nível Superior, Brasil (CAPES), Finance Code 001, Conselho Nacional de Desenvolvimento Científico e Tecnológico (CNPq) (grant 306395/2017-7), Fundação de Amparo à Pesquisa do Estado do Rio Grande do Sul (FAPERGS) (Grants 17/2551-0001197-0 and 19/2551-0001277-2), and Fundação de Amparo à Pesquisa do Estado de São Paulo (FAPESP) (Grant 2020/05165-1).

**Conflicts of Interest:** The authors declare no conflict of interest.

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
