# Peer review of "Multi-Objective Decision-Making Meets Dynamic Shortest Path: Challenges and Prospects†"

_algorithms, doi:10.3390/a16030162_

Round 1

Reviewer 1 Report

The problem raised in the article is essential and current. The authors did an excellent job of reviewing the DSP, MODM, and MOSP problems and the gaps in their methods to solve the MODSP problem. The new thing seems to be a complete formalization of the MODSP problem class and its relationship to the DSP, ASP, SP, and MODM problem classes through a novel taxonomy.

The main weakness of this article is that the above achievement seems insufficient for an unequivocal positive review of this article.
Therefore, I suggest that the authors redraft their article in such a way as to emphasize the novelty of their research. The work should now qualify as a review and not an original piece.

Reviewer 2 Report

The authors present a formal formulation of the Multi-Objective Dynamic Shortest Path (MODSP) problem, which is essentially an extension of the Dynamic Shortest Path problem to a multi-objective environment. As well as a comprehensive review of the literature, some applications are shown. The paper seems to be mathematically sound, and, in my opinion MODSP constitutes a challenging yet useful problem to experts and researchers alike.   

Reviewer 3 Report

The paper provides a survey of the multiobjective dynamic shortest path (MODSP) problem, and its relation with other shortest-path related problems.

The paper is rather shallow, seen as a survey of MODSP, and is missing a huge amount of work on dynamic one-to-all and/or all-to-all shortest path techniques, that should be checked and presented more thoroughly. Moreover, there is a huge amount of work on the (so-called) speedup techniques and oracles, which work as follows: they preprocess distance/travel-related data structures, which they then exploit to provide remarkably fast to arbitrary routing queries. Many of them try also to support remarkably fast (either theoretically, or in practice) updates to the preprocessed information due to unforeseen changes in the graph structure and/or the travel-cost metrics.

The authors are advised to study carefully this quite fruitful line of research, which is already a mature technology for route-planning related problems and is also (at least in some cases) freely provided (source codes) for integration to even commercial systems. 

An excellent starting point (albeit a little bit outdated now), is the following survey:

Hannah Bast, Daniel Delling, Andrew Goldberg, Matthias M¨uller-Hannemann, Thomas Pajor, Peter Sanders, Dorothea Wagner, and Renato F. Werneck: Route Planning in Transportation Networks. In Algorithm Engineering - Selected Results and Surveys, LNCS 9220, pages 19–80. Springer, 2016.

DETAILED COMMENTS FOR AUTHORS

P2, L82, and FIGURE 1: The destination vertex is denoted by $d$ in line 82, but in  line 76 you mention that it is denoted by $\delta$. Please correct both the text in line 82, and Figure 1, or the text in line 76.

P3, L122:  algorithms proportion solutions -->  algorithms propose solutions

P3, L143: fast matrix technic --> fast matrix technique

P3, L144: \tilde{O}(n^{2.045}/e^2) --> \tilde{O}(n^{2.045}/\epsilon^2)

 P3, L144: only direct graphs --> only directed graphs

P3, L153-155: The last sentence "As we have ... our work." does not make sense. Please rephrase it. 

P4, L171-173: The following sentence, trying to define the notion of non-dominated solutions, is inaqurate: "When a solution presents an appropriate compromise between all the objectives and does not degrade any of them concerning all the other possible solutions, we have a non-dominated solution [23]."

Recall that a weakly-non-dominated solution $w$, compared to any other solution $w'$, has either all its objective values at least as good as the objective values of w', or at least one strictly better objective value to that of w. The pareto set (or front) contains all weakly-non-dominated solutions.

P5, L196: Know weights scenario --> The scenario of known weights

P6, L215: need a expert-level --> need expert-level

P7, L273: vertex piked --> vertex picked

P7, L311: their work focus on --> their work focuses on

P8, L324: and allow --> and allows

P8, L337:  in Equation 1, where w represents -->  in Equation 1, represents

P8, L343: Then a path $\pi_{xy)\in G$ cost --> Then, the cost of a path $\pi_{xy)\in G$

P8, L364: returns the approximated Pareto front -->  returns the approximate Pareto front 

NOTE: The exact and approximate pareto sets have not been formally defined in the paper. 

P10, Figure 5: The figure is not quite accurate, SP should probably be a subclass of MOSP, exactly in the same way as DSP is a subclass of MODSP. In particular, as a Venn diagram, the area of MOSP is rather unclear.

P10, L402: class can not solve --> class cannot solve

P11, L449:  so that connection is kept -->  so that connectivity is preserved

Round 2

Reviewer 1 Report

Thank the Authors for applying my suggestions.

I have no more comments or suggestions.